# Effect of Intake of Food Hydrocolloids of Bacterial Origin on the Glycemic Response in Humans: Systematic Review and Narrative Synthesis

**DOI:** 10.3390/nu13072407

**Published:** 2021-07-14

**Authors:** Norah A. Alshammari, Moira A. Taylor, Rebecca Stevenson, Ourania Gouseti, Jaber Alyami, Syahrizal Muttakin, Serafim Bakalis, Alison Lovegrove, Guruprasad P. Aithal, Luca Marciani

**Affiliations:** 1Department of Clinical Nutrition, College of Applied Medical Sciences, Imam Abdulrahman Bin Faisal University, Dammam 31441, Saudi Arabia; noaalshammari@iau.edu.sa; 2Translational Medical Sciences and National Institute for Health Research (NIHR) Nottingham Biomedical Research Centre, Nottingham University Hospitals NHS Trust and University of Nottingham, Nottingham NG7 2UH, UK; Guruprasad.Aithal@nottingham.ac.uk; 3Division of Physiology, Pharmacology and Neuroscience, School of Life Sciences, Queen’s Medical Centre, National Institute for Health Research (NIHR) Nottingham Biomedical Research Centre, Nottingham NG7 2UH, UK; Moira.Taylor@nottingham.ac.uk; 4Precision Imaging Beacon, University of Nottingham, Nottingham NG7 2UH, UK; Rebecca.Stevenson@nottingham.ac.uk; 5Department of Food Science, University of Copenhagen, DK-1958 Copenhagen, Denmark; ourania@food.ku.dk (O.G.); bakalis@food.ku.dk (S.B.); 6Department of Diagnostic Radiology, Faculty of Applied Medical Science, King Abdulaziz University (KAU), Jeddah 21589, Saudi Arabia; jhalyami@kau.edu.sa; 7Indonesian Agency for Agricultural Research and Development, Jakarta 12540, Indonesia; smuttakin@gmail.com; 8School of Chemical Engineering, University of Birmingham, Birmingham B15 2TT, UK; 9Rothamsted Research, Harpenden, Hertfordshire AL5 2JQ, UK; alison.lovegrove@rothamsted.ac.uk

**Keywords:** glycemic response, blood glucose, bacterial, polysaccharides, gums, satiety, xanthan, pullulan, dextran

## Abstract

Diabetes mellitus is a chronic condition characterized by increased blood glucose levels from dysfunctional carbohydrate metabolism. Dietary intervention can help to prevent and manage the disease. Food hydrocolloids have been shown to have favorable properties in relation to glycaemic regulation. However, the use of food hydrocolloids of bacterial origin to modulate glucose responses is much less explored than other types of hydrocolloids. We, therefore, carried out the first review examining the impact of intake of food hydrocolloids of bacterial origin (as a direct supplement or incorporated into foods) on glycemic response in humans. Fourteen studies met the inclusion criteria. They used either xanthan gum, pullulan, or dextran as interventions. There was a wide variation in the amount of hydrocolloid supplementation provided and methods of preparation. Postprandial blood glucose responses were reduced in half of the studies, particularly at higher intake levels and longer chain hydrocolloids. When xanthan gum was added to the cooking process of muffins and rice, a significant reduction in postprandial blood glucose was observed. The use of these hydrocolloids is potentially effective though more research is needed in this area.

## 1. Introduction

Diabetes mellitus is a chronic condition characterized by increased circulating blood glucose levels resulting from dysfunctional carbohydrate metabolism. Age, obesity, and physical inactivity are all factors reported to increase the risk of the disease, particularly in people who are genetically susceptible [1]. Type 2 diabetes mellitus (T2DM) has the highest incidence, accounting for at least 90% of all the cases of diabetes globally. Currently, about 387 million people are reported to have this condition, and an estimated 179 million people are undiagnosed. On average, approximately 2.2 million deaths are directly caused by diabetes annually [2]. Estimates from the International Diabetes Federation show that by 2035, more than 592 million people will have T2DM globally, and most of them (about 77%) will be living in low- and middle-income countries. The management of elevated glucose levels in T2DM includes interventions designed to increase physical activity and dietary changes.

The glycemic index (GI) is a ranking system to reflect how a specific food will increase blood glucose level, usually within two hours after consuming that food compared to white bread or glucose equivalent carbohydrate load set to a value of 100. Accordingly, foods with low GI result in lower postprandial glucose levels and lower postprandial insulin levels [3]. Reducing postprandial glucose excursions may help reduce the risk of T2DM complications [4]. To this effect, hydrocolloids have long been known to have a lowering effect on postprandial hyperglycemia since early work on guar gum [5,6,7] and gelatin [8].

Hydrocolloids gums are polysaccharides used to increase viscosity and gelatinization and have many applications in the food industry as thickeners, emulsifiers, and stabilizers. Hydrocolloids slow starch digestion potentially due to the formation of a physical barrier that limits the effect of digestive enzymes [9,10,11]. This can be attributed to the inhibitory effect of the hydrocolloid on protein hydrolysis to immediate binding of the hydrocolloids to enzymes, therefore slowing the interaction between the enzyme and the substrate. It has also been suggested that hydrocolloids may alter enzymes conformation [11,12]. The addition of hydrocolloids gums through food processing can also reduce the overall digestibility of foods [13]. Hydrocolloids can reduce or restrict contact between enzymes and substrates, reduce mass transfer in the gut (e.g., by increasing viscosity or gelling) and inhibit enzyme activity [14,15,16].

Hydrocolloids can be categorized depending on their function as thickening agents (xanthan gum, locust bean gum or LBG, carboxymethyl cellulose or CMC and guar gum) or gel-forming agents (agar-agar or AA, carrageenan, gelatin, pectin, gellan gum, and konjac-glucomannan or KG). It is also useful to characterize hydrocolloids based on their natural source: (1) plants, (2) seeds, (3) animal extract, and (4) bacterial or microbial polysaccharides [17].

While other hydrocolloids have been investigated extensively and their effect on glycemic control reviewed, there has been little focus on hydrocolloids from the microbial origin (also known as bacterial polysaccharides or microbial polysaccharides) in relation to glycemic control. Bacterial or microbial polysaccharides gums are polysaccharides produced extracellularly by microorganisms for different purposes in the food, pharmaceutical, and medical industries. The common hydrocolloids of bacterial origin include xylinan, scleroglucan, schizophyllan, and xanthan, gellan, curdlan, pullulan, and dextran. Xanthan gum (XG) is a polysaccharide by-product of the bacteria Xanthomonas campestris. It consists of repeated units of glucose, mannose, and glucuronic acid (Figure 1A). Discovered in 1968, it has a range of food uses approved by the Food and Drug Administration, including its use as a thickening agent and stabilizer. Pullulan is a polysaccharide polymer consisting of maltotriose as a building block (Figure 1B). Three glucose units of maltotriose are linked through α-(1→4) glycosidic bonds [18]. It is produced by the fungus Aureobasidium pullulans and is used in food as a texturizer. Dextran is a complex branched polysaccharide (Figure 1C) produced from lactic acid bacteria.

This systematic review aimed to investigate and examine the effect of hydrocolloids of bacterial origin (as a direct supplement or incorporated into foods) to improve glycemic control and appetite in healthy individuals and those diagnosed with type 2 diabetes. Tolerance issues reported in the selected literature will also be noted.

## 2. Materials and Methods

The review protocol was informed by the guidelines of the Preferred Reporting Items for Systematic Review and Meta-Analysis (PRISMA) [19]. The review protocol is available at PROSPERO with registration number CRD42021227056 and can be accessed at PROSPERO (https://www.crd.york.ac.uk/prospero/). (accessed on 8 June 2021).

### 2.1. Inclusion and Exclusion

The inclusion criteria were based on the Participant–Intervention–Comparator–Outcomes–Study design (PICOS) format. Participants: studies were considered for inclusion only if all participants were adults aged 18 or above, male and female, and were healthy, prediabetic, or diagnosed with diabetes. All study participants that met the criteria were included, regardless of ethnicity and gender. Any studies on animal models and in-vitro were excluded. Intervention: only studies in which the intervention included diet consumption of any type of hydrocolloids from bacterial origin were included: xanthan, gellan, pullulan, dextran, curdlan, scleroglucan, schizophyllan, or cyanobacterial; any studies that used a different type of dietary intervention were excluded. Comparison: no intervention or comparator group, placebo, standard food intake. Outcomes: the primary outcomes of interest included demonstrating the effectiveness of hydrocolloid gum in modifying blood glucose response, fasting and postprandial glucose, and insulin. Secondary outcomes are appetite and satiating effect. Any adverse events associated with the intervention were also noted, as this indicated safety and tolerance. Study design: randomized controlled trials, quasi-experimental studies (non-randomized controlled trials, before-and-after,) with the last search completed on 4 February 2021. Only studies published in English were included. Any studies using secondary data (such as reviews) were excluded, as were non-experimental research, ongoing trials, and meeting abstracts.

### 2.2. Search Strategy

An electronic search was performed using 3 databases, Web of Science, MEDLINE, and Embase, between 1950 and 2021. Some databases were available only from 1974 onwards. An initial search was conducted in late 2020 and then updated on 4 February 2021. An advanced literature search was performed by using “AND” and “OR” operators to narrow down or expand its scope, increasing retrieval effectiveness. Along with Boolean search techniques, references from eligible journal articles were trawled to identify as many primary sources as possible. The search keywords were not restricted to just the microbial polysaccharides to avoid the risk of missing papers focused mainly on some of the more common hydrocolloids. Using synonyms and alternating search terms significantly increased the number of hits or search results. Reference chaining and manual searches on other engines such as Google and Google Scholar were also carried out to capture possible hits missed by the initial electronic search on the main databases. All the retrieved references by the electronic databases were exported to an EndNote database.

The search strategy and keywords were set as follows:(hydrocolloid* or polysaccharid*).ti,ab.(xanthan or xylinan or acetan or “hyaluronic acid” or gellan or curdlan or pullulan or dextran or scleroglucan or schizophyllan or cyanobacterial).ti,ab exp hydrocolloid/1 or 2(“glyc?emic index*” or “glyc?emic response*” or “glyc?emic load*”).ti,ab.glycemic index/glycemic load/4 or 5 or 6((glucose or sugar) adj2 blood).ti,ab.((glucose or sugar) adj2 plasma).ti,ab.Blood Glucose/8 or 9 or 107 or 113 and 12limit 13 to human

### 2.3. Study Selection, Data Extraction, and Synthesis

Two researchers (N.A. and L.M.) independently screened the articles by evaluating titles and abstracts of studies retrieved from the electronic database against the inclusion and exclusion criteria. After screening, all the relevant studies were retrieved as full-text articles. Ethical approval was not required, as only published studies were included in the analysis.

The titles, abstracts, and content of these articles were screened to ensure that they were relevant to the current study using a data extraction form. Information about the participants, interventions, comparisons outcomes, study design (PICOS), and other details such as year of publication, research setting, main findings, study limitations, and follow-up were documented. The primary outcomes of interest included demonstrating the effectiveness of hydrocolloid gum in improving the glycemic profile. Adverse events and adherence were also noted as indicators of safety and tolerance.

Statistical analysis was not undertaken as the methods and outcomes of the selected studies were heterogeneous. As such, a narrative synthesis of the included studies was used to answer the research question.

### 2.4. Risk of Bias across Studies

Each included study was assessed for potential risk of bias (for example, from the study design and reporting) by two researchers independently (N.A. and L.M.) using the ROBINS-I tool [20,21]. Seven domains of bias were evaluated. The pre-intervention domains were biased due to confounding and bias in the selection of participants into the study. The intervention bias domain was biased in the classification of interventions. The post-intervention domains were biased due to deviations from intended interventions, bias due to missing data, bias in the measurement of outcomes, and bias in the selection of the reported result. Each domain consisted of 3–8 signaling questions. If the signaling questions in every domain were answered with “No/Probably No”, the study was considered to have a low risk of bias, and no further signaling questions were considered. If one or more questions were answered by “Yes/Probably Yes”, this was considered a potential marker for risk of bias, and further questions related to this domain were assessed. Any disagreement was resolved through discussion. Finally, data were collected from each study and extracted into a Microsoft Excel spreadsheet. The risk of bias figure was created using the ROBINS-I tool.

## 3. Results

The electronic search identified 1327 records from the databases (see review flow diagram in Figure 1), including 209 from Medline, 430 from EMBASE, and 688 from Web of Science. One hundred forty-five papers were found to be duplicates, leaving 1182 citations for initial screening. Three papers were added from cross-referencing and other internet searches. This yielded 50 records for full-text screening. Fourteen papers satisfied the criteria and were included in this systematic review. Figure 2 summarizes the process of literature search, identification, and screening, based on the PRISMA flow chart. Of these 14 papers, 4 investigated pullulan [22,23,24,25], 7 investigated xanthan gum [26,27,28,29,30,31,32] and three investigated dextran [33,34,35].

The characteristics and main findings of the included studies are summarized in Table 1 and described in more detail below.

### 3.1. Pullulan and Glycemic Response

Wolf et al. [22] reported a randomized, crossover, double-masked, two-period, two-treatment study of a 474 mL juice-like beverage containing 50 g of carbohydrate from pullulan or maltodextrin control. They measured finger-prick blood glucose at 15, 30, 45, 60, 90, 120, 150, and 180 min postprandially. Twenty-eight participants completed the study. Compared with control, incremental peak blood glucose concentration was significantly lower when subjects consumed pullulan (4.24 ± 0.35 mmol/L versus 1.97 ± 0.10 mmol/L; *p* < 0.01). Pullulan had a significantly lower incremental area under the glucose curve by 50%.

In the study by Spears et al. [23], 50 g of low-molecular-weight pullulan was provided in 474 mL juice-like beverage and compared to 50 g maltodextrin control in 34 healthy participants. The study day consisted of a 3-h tolerance test measuring finger-prick capillary plasma glucose and serum insulin. These data showed that low–molecular weight pullulan did not reduce the incremental plasma glucose response compared to maltodextrin. Late phase plasma glucose concentration was significantly higher for pullulan at 150 and 180 min 216 ± 21 compared to control 197 ± 25 (*p* < 0.05). However, the postprandial incremental serum insulin response was reduced by pullulan, 237 ± 25 compared to control 309 ± 30. The incremental peak serum insulin was significantly reduced by 30% when subjects consumed pullulan compared to maltodextrin.

Based on ROBINS-I tool for assessing the risk of bias, 11 studies were considered as low risk of bias, one as a moderate and 2 as high risk. These results are presented as a summary plot in Figure 3.

Peters and colleagues [25] carried out a randomized controlled study in 35 participants who were given a 325 mL commercial meal replacement drink containing 15 g medium-chain pullulan, long-chain pullulan, or maltodextrin as control. A subset of 12 participants underwent glucose measurements. The area under the curve for glucose of the two pullulan drinks was significantly higher than that for maltodextrin. The latter showed the greatest blood glucose excursion, with the medium-chain pullulan exhibiting an intermediate excursion and the long-chain pullulan the lowest excursion.

The study by Stewart et al. [24] provided a sustained fiber intervention with 12 g/day (divided into two daily doses) taken mixed with a commercial apple sauce for 14 days. The pullulan was used, and it was compared with resistant starch, soluble fiber dextrin, soluble corn fiber, and maltodextrin as a control. Only fasting glucose was measured after the intervention, and this did not differ from the control.

### 3.2. Xanthan and Glycemic Response

Osilesi et al. [26] conducted a crossover trial, nine borderline Type 2 diabetic subjects and four healthy controls that received six muffins containing 12 g xanthan gum or control for 6 weeks for each (consecutive) arm of the study in an otherwise free-living setting. The addition of xanthan gum significantly lowered fasting glucose 76 ± 3 compared to control 91 ± 3 and post-load serum glucose 98 ± 3 compared to 131 ± 15 after the first 6 weeks (*p* < 0.01). In the diabetic group, the xanthan containing muffin intervention lowered fasting glucose by 38% compared to control muffin intervention (93 ± 10, 149 ± 13) (*p* < 0.05) and post-load glucose by 31% (*p* < 0.05).

A study by Eastwood et al. [27] investigated daily doses of xanthan varied between 10.4 g and 12.9 g over 23 days. In this before and after study design, the results showed no significant effect on blood glucose, with a small increase at 30 min after glucose tolerance test ingestion of xanthan.

A study by Edwards and colleagues [28] assessed xanthan. On sixteen healthy participants received a 250 mL drink containing a 50 g glucose load and 2.5 g of viscous polysaccharide (xanthan or xanthan/locust bean gum or xanthan/Mey or locust bean gum or guar). Glucose was significantly reduced by the addition of xanthan, compared with control. The area under the curve for glucose at 2 h was significantly lower for the xanthan group (129 ± 23, 217 ± 21 respectively, *p* < 0.05).

In a randomized controlled trial by Paquin et al. [29], 14 healthy male subjects consumed four 300 mL juice drinks on four different study days, at least one week apart. One was a control juice. The incremental peak of glucose showed a significant reduction in the mix of xanthan gum and β-glucan compared to control (*p* = 0.001). The IAUC for glucose, insulin overall 2 h test showed no significant overall treatment effect.

The study by Tanaka et al. [31] explored the effects of 1% *w*/*v* xanthan gum added to 150 mL of a commercial semi-solidified enteral feed versus the same enteral feed without xanthan gum as a control on the blood glucose level. Five healthy participants drank the control feed on a morning study day after a 12 h fast. They then repeated the study after 7 days washout drinking the enteral feed plus xanthan gum intervention. The results showed that blood glucose levels were 22% lower for the xanthan gum intervention compared to control at 20 min postprandially (*p* < 0.05) and overall, as delta area under the curve for 120 min (*p* < 0.05).

In Fuwa et al. [30], the xanthan was mixed into the rice meal and added directly during the rice cooking. The concentrations used were 0, 0.5, 1.0, and 2.5% of raw rice weight. The participants underwent a 12 h fast before the studies. The addition of ≥1.0% xanthan gum during rice cooking significantly reduced blood sugar levels at 15 and 30 min, and this was still significantly lower at 45 min for participants who received the meal with xanthan at 1.5%. Blood glucose was also significantly reduced by adding the xanthan sol at 15–60 min postprandially. The 15-min postprandial blood sugar levels were significantly reduced in subjects who had consumed the sol form of xanthan gum before rice. The findings suggested that the blood sugar levels after rice consumption were suppressed most effectively when the rice was coated in xanthan gum sol.

In a recent publication, Naharudin et al. [32] used xanthan gum in three arm-study. The first intervention was water control, and the second intervention a viscous breakfast (eaten with a spoon) based on low-calorie orange squash and xanthan gum. The third breakfast was the same second intervention but with about 20 times more energy added as maltodextrin. For the purpose of this review, the relevant sample is the second one, the low-calorie orange squash and xanthan gum which, compared to water control, did not alter blood glucose.

### 3.3. Dextran and Glycemic Response

Two early studies used dextran as a feeding intervention. One [33] gave 100 mL of 20% dextran to two fasted human participants and 200 mL of 20% dextran to another two fasted participants. They observed a modest rise in blood sugar over 2 h post-intervention and concluded that intestinal breakdown of the dextran into glucose was possible and seemed rapid. The other study [34] delivered 500 mL of 5% un-hydrolyzed high molecular weight dextran orally to five patients. It did not observe any significant rise in blood sugar in these participants over 3 h postprandial.

Subhan et al. [35] reported two trials investigating the effects of ingestion of isomalto-oligosaccharides. In trial 1, dextran was used as a comparator/placebo. In the trial, 20 g of dextran, isomalto-oligosaccharides, maltodextrin, dextrose, and water were given to 12 healthy participants. Dextran did not increase blood glucose compared with water. The treatments and the data showed that dextran’s values were both significantly lower than for the isomalto-oligosaccharides, noting that the Glycemic Glucose Equivalent for dextran was 0.28 ± 0.08, much lower than 1.

These were small-scale studies of dextran that were not followed up in the intervening years.

### 3.4. Appetite

Two studies examined the effect of pullulan intake on appetite. Stewart et al. [24] assessed hunger using visual analog scales (VAS) and measured before breakfast, lunch, and dinner on day 3 and day 14, and no differences were found between treatments and control. Peters et al. [25] assessed appetite using VAS, and appetite scores were significantly reduced for the long-chain pullulan drink compared to maltodextrin, whilst the medium-chain pullulan was different from maltodextrin.

Three studies examined the effect of xanthan gum consumption on appetite. In the study by Osilesi et al. (1985) [26], participants reported a sense of fullness after consuming the xanthan-containing muffin. Paquin et al. [29] measured appetite by (VAS) questionnaires. There was no significant overall effect on satiety for 2 h test. Naharudin et al. [32] assessed the hunger using VAS reported that xanthan gum lowered hunger and increased fullness compared to control. They concluded that the increased viscosity caused by the xanthan was responsible for the changes in satiety.

### 3.5. Tolerance

Four studies examined the overall tolerance of pullulan consumption. Wolf et al. [22] assessed gastrointestinal tolerance using questionnaires. The frequency and intensity of flatulence were significantly higher after subjects consumed pullulan compared with control. Spears et al. [23] assessed malabsorption and gastrointestinal tolerance using symptom questionnaires for 24 h after consuming the test meal. Gastrointestinal symptoms were generally of low intensity. Peters et al. [25] reported a low gastrointestinal disturbance for the pullulan group. Stewart et al. [24] reported that gastrointestinal symptoms were moderate albeit significantly higher with pullulan.

Two studies noted the tolerance of xanthan gum. Osilesi et al. [26] reported that the intervention was well tolerated with no side effects. Eastwood et al. [27] reported no sign of adverse effects from the ingestion of xanthan at that dose.

## 4. Discussion

This systematic review investigated the evidence related to the dietary intake of food hydrocolloids of bacterial origin in improving the glycemic response in humans. This category of food hydrocolloids is still under-researched compared with other types of hydrocolloids. Our searches found 14 publications on three food hydrocolloids of bacterial origin: pullulan, xanthan, and dextran.

Pullulan reduced blood glucose excursions depending on its chain length and digestibility, the effect being stronger for longer-chain, lower digestibility pullulan [25]. This was shown for a 15 g acute supplementation. The positive effect on reducing glycemic excursion was confirmed for pullulan that is hydrolyzed slowly over time. At a higher acute dose (50 g), information on the chain length was not provided [22]. Pullulan with low-molecular-weight can be rapidly hydrolyzed in the small bowel and could not reduce the glycemic response compared with maltodextrin even at the relatively high acute dose (50 g) [23]. An additional paper studied a low (5 g) acute dose of pullulan, but this was given in combination with other materials (effective by themselves in reducing glucose responses, such as resistant starch). Therefore, it was not possible to isolate the effect of pullulan [36]. In the only chronic study providing a 12 g pullulan intervention daily for 2 weeks, there was no effect of pullulan on fasting glucose. However, postprandial responses/areas under the curve for glucose were not measured in this case [24]. Not all the pullulan literature had data on satiety, with the only positive effect on reducing appetite shown by the long-chain type [25]. Slow digestibility and malabsorption result in some of the pullulan reaching the colon and fermentation. This causes some gastrointestinal symptoms such as flatulence, which may be a tolerance issue, particularly at high doses or repeated dosing, especially in those with particular sensitivity.

Xanthan ingestion was generally well-tolerated, but the results on blood glucose responses are more mixed. This hydrocolloid showed a positive effect in reducing blood glucose responses in a group of diabetic and healthy participants at repeated dosing of 12 g per day for 6 weeks [26]. However, this was not observed in healthy participants after 23 days of repeated ingestion of around 10–13 g daily [27]. The addition of 2.5 g of xanthan to an acute glucose load drink reduced the area under the curve for glucose [28]. However, an addition of about 0.5 g to an acute intervention juice drink did not change the area under the curve for glucose [29] nor in addition to a viscous breakfast [32].

Interestingly the addition of 1% *w*/*v* xanthan gum to enteral feeding in an acute setting lowered blood glucose levels [31]. An additional report studied a low acute dose of xanthan in a drink, but this was given in combination with other materials (guar gum or konjac mannan). Therefore it was not possible to isolate the effect of xanthan [37]. Only some of the papers on xanthan considered satiety. Some positive effects were shown [32], possibly only due to the increase in viscosity, although another study showed no significant effects [29]. There was again limited information for the secondary outcome of satiety. One of the studies on juices showed a positive effect of xanthan on satiety [32], while in another study, there was little effect compared to the control [29]. In one of the studies, the participants reported (against no formal measurement) an occasional sense of fullness following the intervention and no gastrointestinal symptoms [26].

One of the most interesting findings of this review was the effect of modifying food processing during cooking using the xanthan hydrocolloid. Adding xanthan gum to the muffins reduced fasting glucose significantly [26]. Additionally, [30] added xanthan gum to rice cooking, resulting in a significant reduction in blood glucose responses. This is an interesting and relatively simple intervention to modulate glucose responses effectively by coating a typically high glycemic index staple food (rice) with a food hydrocolloid and warrants further investigation.

There is much less data on dextran. Early work indicated that it had modest to no effects on blood glucose for 2–3 h post-prandially, whilst a recent study showed that 20 g dextran did not increase blood glucose compared to control.

The mechanisms whereby food hydrocolloids may modulate blood glucose responses include surface interactions acting as a barrier for enzymatic access, restricting leakage of amylose chains during gelatinization, and increasing digesta viscosity. This may slow the release and breakdown of nutrients from the food matrix into absorbable forms [38,39].

Some of the limitations highlighted by this review are that these studies used markedly different doses of hydrocolloids, from small to what may be considered high amounts. Xanthan, pullulan, and dextran are considered safe for dietary consumption. The European Food Safety Authority EFSA re-evaluated xanthan gum in 2017 and concluded that it was safe for use without any need to set a maximum dose for intake [40]. Most of the studies included in our review were carried out using one-off, acute doses monitored for a short period. Sustained exposure to food interventions may elicit body adaptation and provide different responses [41]. Many of the studies included interventions using small numbers of participants and often only healthy volunteers. Moreover, the data are generally reported without comparison between male and female participants; therefore, it was not possible to comment on possible gender differences in the glycemic effect of these hydrocolloids of bacterial origin. It was also difficult to categorize the risk of bias of the two older studies [33,34] using the ROBINS-I tool due to the lack of information provided in those short reports.

## 5. Conclusions

The results from this systematic review indicated a mixed picture overall, with only half of the studies reporting a reduction in blood glucose responses upon intervention with food hydrocolloids of bacterial origin. The amounts of hydrocolloids provided and the methods varied substantially between studies, and only three of the hydrocolloid types have been studied, with the others remaining to be investigated. Further work with larger numbers of subjects, both with acute and longer-term interventions, is needed. Modifying food processing using food hydrocolloids of bacterial origin may be a promising strategy to help modulate glucose excursions.

## Figures and Tables

**Figure 1 nutrients-13-02407-f001:**
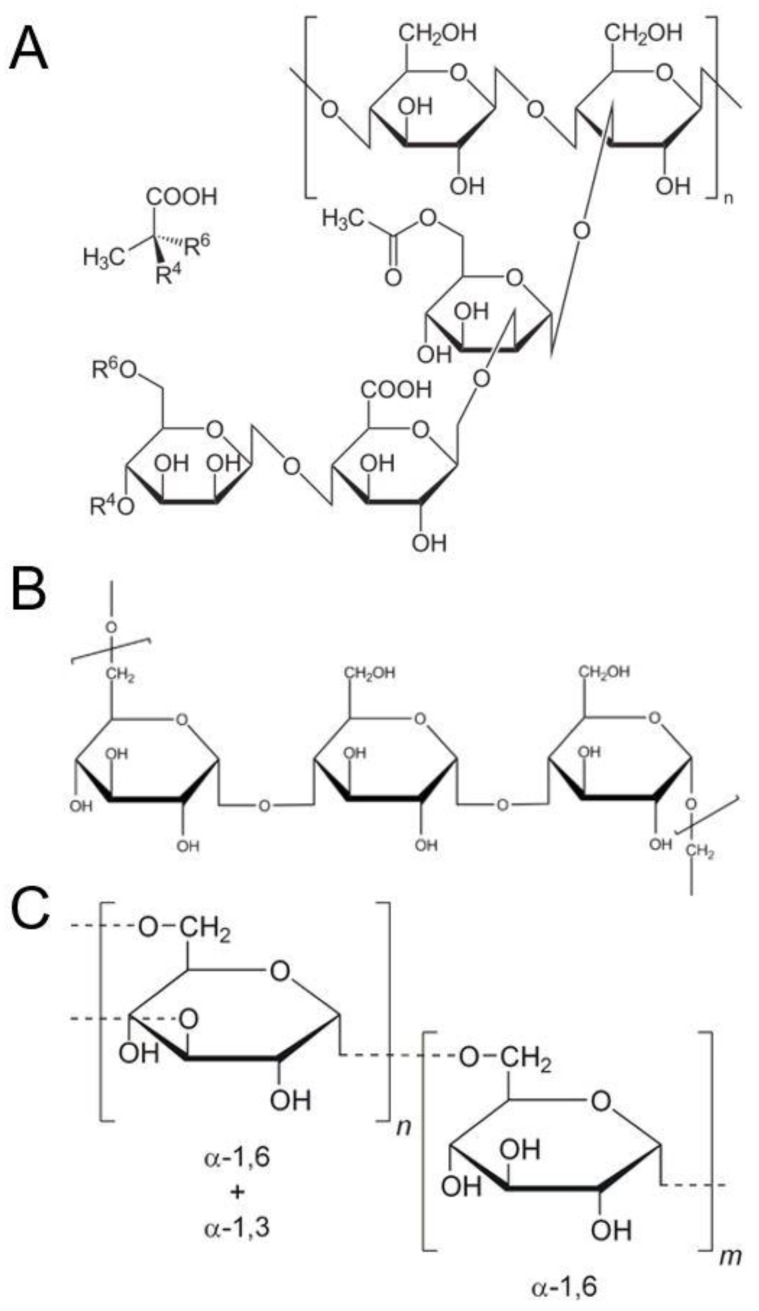
Chemical structures of (**A**) xanthan, (**B**) pullulan, and (**C**) dextran. These structures are in the public domain and are reproduced from a copyright-free repository.

**Figure 2 nutrients-13-02407-f002:**
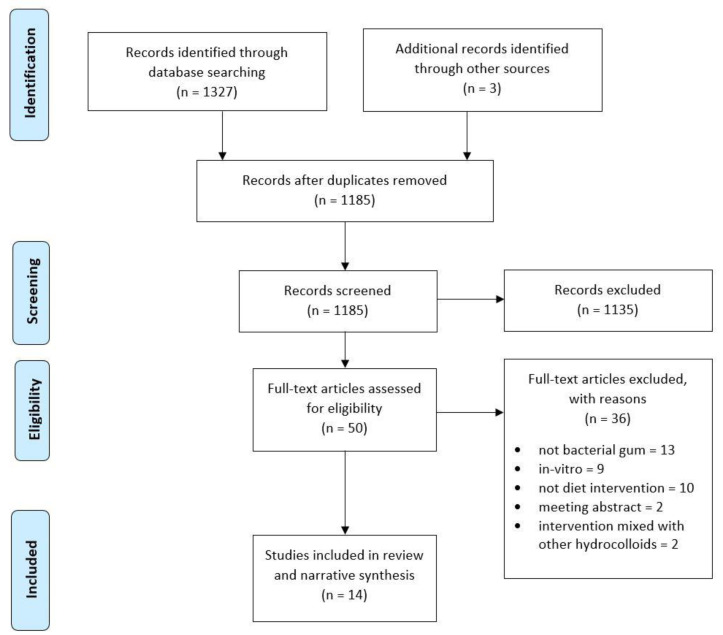
PRISMA flow diagram of Study selection [19].

**Figure 3 nutrients-13-02407-f003:**
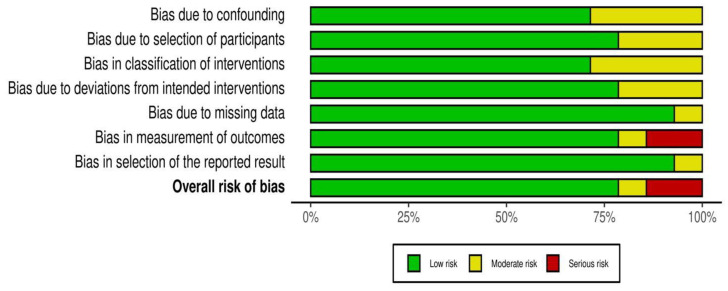
Summary plot of the overall risk of bias.

**Table 1 nutrients-13-02407-t001:** Summary characteristics and results of the included studies.

Author Year Country	Study Design	Intervention Acute/Long-Term and Details	Comparator	Duration of Glucose Assessment	Washout	*n*	Gender	Age	Glucose Results	Appetite Results	Other Outcomes Reported
Bloom 1952 USA	One-way	Acute intervention 100 mL of 20% dextran (*n* = 2); 200 mL of 20% dextran (*n* = 2)	N/A	2 h	N/A	4	N/K	N/K	Low rise in blood sugar over 2 h	N/A	N/A
Joelson 1956 USA	One-way	Acute intervention 400 mL of 5% dextran	N/A	3 h	N/A	5	N/K	N/K	No increase in postprandial blood sugar	N/A	N/A
Osilesi 1985 USA	Crossover	Long-term intervention Xanthan containing muffins 12 g/day for 6 weeks	xanthan-free muffins for 6 weeks	2 h	no wash out	9 DP, 4 HV	M = 2,F = 7 M = 1,F = 3	53 ± 4 37 ± 5	Prior feeding of xanthan induced reduction in post-load glucose by 31% in patients and 25% in HV	Episodic increased fullness no gut symptoms	Reduction in fasting glucose and total cholesterol in both groups. No significant change in insulin, gastrin, and GIP, and triglycerides in patients
Eastwood 1987 UK	Before-after	Long-term intervention Xanthan as fluid gel three times daily for 23 days, total between 10.4 and 12.9 g/day	N/A	4 h	N/A	5 HV	M = 5	26–50	No significant effect on plasma glucose	N/A	Increased fecal weight and intestinal transit. No effect on plasma biochemistry, hematological indices, urinalysis, insulin, serum immunoglobulins, triglycerides, phospholipids and cholesterol
Edwards 1987 UK	Randomized crossover	Acute intervention 50 g glucose drink and (1) 2.5 g xanthan; (2) xanthan and locust bean gum; (3) xanthan/Mey; (4) locust bean gum or guar	50 g glucose control drink	2 h	N/K	16 HV	M = 12 F = 4	18–25	Significant reduction in glucose AUC	N/A	Reduction in insulin AUC. No change in gastric emptying
Wolf 2003 USA	Randomized crossover	Acute intervention 474 mL beverage with pullulan 0.1 g/1 mL = 47 g	Maltodextrin beverage	3 h	5–13 days	28 out of 36 HV	M = 22 F = 14	18–75	Positive incremental AUC reduced by 50%	N/A	Increased breath hydrogen concentration
Spears 2005 USA	Randomized crossover	Acute intervention beverage with low molecular weight pullulan 50 g	Maltodextrin beverage	3 h	4–14 days	34 HV	M = 19 F = 15	20–39	No effect on incremental plasma glucose response	N/A	Higher breath hydrogen at later time points. Serum insulin lower during the first 90 min postprandially, higher at 3 h. No effect on symptoms
Stewart 2010 USA	Single-blind randomized crossover	Long-term intervention Sauce mixed with 12 g/day of (1) pullulan; (2) resistant starch; (3) soluble fiber; (4) soluble corn fiber give for 14 days	Maltodextrin	Only fasting time point	21 days	20 HV	M = 10 F = 10	32 ± 5 38 ± 4	No significant change in fasting glucose	No change in hunger	Increase in gastrointestinal symptoms. No change in ad libitum diet, stool parameters, trigycerides, cholesterol, insulin, C-reactive protein, ghrelin, blood pressure and body weight
Peters 2011 Netherland	Randomized crossover	Acute intervention Drink containing 15 g pullulan as (1) long-chain; (2) medium-chain	Maltodextrin drink	5 h	week	35 HV	F = 27 M = 8	20–59	Subset of *n* = 12 tested for glucose. Significant increase of AUC 0–150 min in medium-chain and long-chain	Significant reduction in long-chain group 0–150 min	Breath hydrogen was significantly higher for both chain lengths. Ony occasional complaints of symptoms. Insulin was lower for long-chain pullulan
Paquin 2013 Canada	Randomized crossover	Acute intervention 4 juices 300 mL: (1) xanthan 0.18 g/100 mL; (2) B-glucan; (3) xanthan + B-glucan 0.09 g/100 mL; (4) control	Control juice	2 h	1 week	14 HV	M	20–50	No difference in AUC compared to control group	No change	No significant change in insulin
Fuwa 2016 Japan	Crossover	Acute intervention 250 g Rice with: (1) xanthan added during rice cooking 0.5, 1.0, 1.5, 2.0, and 2.5%; (2) xanthan mixed with cooked rice at 0.5, 1.0, 1.5, 2.0, and 2.5%	250 g cooked rice	2 h	>3 days	11 HV	F	19–39	Significant decrease in postprandial glucose at 15, 30 and 45 min in >1, 1.5% added during rice cooking group and in xanthan sol group at 15–60 min	N/A	N/A
Tanaka 2018 Japan	Ordered intervention	Acute intervention 150 mL enteral formula + 1.0% xanthan	150 mL enteral formula	2 h	7 days	5 HV	M = 2 F = 3	21–22	AUC glucose 48% smaller than control	N/A	N/A
Naharudin 2020 UK	Randomized double blind	Acute intervention Semi-solid fasts containing xanthan 0.1 g/kg of body weight. One with and one without added carbohydrate	Water as control	105 min	>4 days	22	M = 22	23 ± 3	No significant change in plasma glucose	Significant reduction in hunger, increase in fullness compared to control	Significant differences in insulin and ghrelin in the added carbohydrate group
Subhan 2020 Canada	Trial 1. Single-blind randomized crossover	Acute intervention 20 g of: (1) isomalto-oligosaccharides; (2) dextran; (3) maltodextrin; (4) dextrose reference	Water as control	2 h	1 week	12 HV	N/K	18–75	Dextran did not increase plasma glucose compared with water	N/A	N/A

AUC = area under the curve, DP = diabetic patient, M = male, F = female, HV = healthy volunteer, N/A = not applicable, N/K = not known.

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
