# Peer review of "Effect of Intake of Food Hydrocolloids of Bacterial Origin on the Glycemic Response in Humans: Systematic Review and Narrative Synthesis"

_nutrients, 2021, doi:10.3390/nu13072407_

Round 1

Reviewer 1 Report

The authors have addressed all of my concerns. I have no further comments.

Reviewer 2 Report

The authors addressed the comments and included them in the text.

Reviewer 3 Report

The manuscript submitted for evaluation is a valuable and well-prepared systematic review of the effects of hydro-colloids of bacterial origin on appetite and glycemic control in healthy subjects and subjects with DM2. The work has clear and transparent inclusion criteria and meets a large part of the AMSTAR criteria for a good systematic review. However, I am not sure whether 2 independent researchers conducted database searches or only "study selection" and "study extraction" from already retrieved? This information should be clearly provided in. In my opinion, the work meets the requirements of the publishing house and is a valuable study that can be accepted for publication in its current form.

This manuscript is a resubmission of an earlier submission. The following is a list of the peer review reports and author responses from that submission.

Round 1

Reviewer 1 Report

MAJOR:

1. I am concerned that the selection and outcome of studies are not correctly reported according to the stated inclusion/exclusion criteria.  I only checked one study (Paquin, 2013), but this completely broke all the stated rules.  It is stated that only hydrocolloids of bacterial origin (listed in lines 113-14) are included and that "any studies that used a different type of dietary intervention were excluded."  Paquin tested the acute effect on postprandial glucose and insulin of xanthan gum (XG) alone, beta-glucan (BG) alone and a XG/BG mixture.  Only BG alone and the XG/BG mixture significantly lowered peak glucose; XG alone did not.  Yet table 3 indicates there was a significant decrease in peak glucose.  The text indicates that the XG/BG mixture reduced glucose, but that treatment should have been excluded.  XG alone had no significant effect on any endpoint and, thus, the table should have indicated no significant effect.  This calls into question the ability of the authors to accurately report the results according to their stated criteria, and, therefore, the veracity of the any of the results.

2) The outcome for all the major relevant predetermined endpoints should to be summarized in the table.  For example, Paquin also reported glucose AUC and insulin peak and AUC, but these results are not shown in the table (although they are noted in the text).

3. The results of acute and longer-term studies do not have the same meaning, and it is not clear what kind of study the authors are describing.  Acute test meals studies measuring postprandial responses should be presented separately from other types of study and the nature of the endpoints described clearly indicated.

4. In table 3 study duration is presented in a misleading fashion.  For acute postprandial studies, the relevant factor is not how long it took to complete the study, but how long the assessment of the effect of the intervention lasted.  Thus, for Paquin, the duration is not 1 week as stated; the relevant duration is 2hr (the time over which glucose response was measured).

Because of these major issues, it is not possible to comment on the Results or Discussion sections.

Reviewer 2 Report

This review describes the use of food hydrocolloids of bacterial origin and their effect on post-prandial glucose responses in humans. The manuscript was well written and the inclusion, exclusion criteria for the study are clearly explained.

Minor comment:

  1. Authors summarized the effect of different hydrocolloids on post- prandial glucose response in human subjects, however there was no comparative description in case of males and females. Does hydrocolloids have any differential effects on post-prandial glucose responses in females or males? Please comment on it.

Reviewer 3 Report

This review summaries the research findings on the glycemic responses of humans to the ingestion of food hydrocolloids of bacterial origin. This manuscript was clearly written and easy to follow. The figures and table were well prepared to help readers understand the manuscript. I have a couple of minor comments:

(1) Shouldn’t it be glycemic response rather than glycaemic response?

(2) In line 188, what does figure 1 in parenthesis mean? Is it related to Figure 1?